# DRUGPATH: The Drug Gene Pathway Meta-Database

**DOI:** 10.3390/ijms21093171

**Published:** 2020-04-30

**Authors:** Rajeev Jaundoo, Travis J. A. Craddock

**Affiliations:** 1Department of Biomedical Engineering, University of Alberta, Edmonton, AB T6G 2R3, Canada; jaundoo@ualberta.ca; 2Institute for Neuro-Immune Medicine, Nova Southeastern University, Fort Lauderdale, FL 33313, USA; 3Departments of Psychology and Neuroscience, Computer Science, and Clinical Immunology, Nova Southeastern University, Fort Lauderdale, FL 33313, USA

**Keywords:** polypharmacology, database, gene, drug, target, combination therapy

## Abstract

The complexity of modern-day diseases often requires drug treatment therapies consisting of multiple pharmaceutical interventions, which can lead to adverse drug reactions for patients. *A priori* prediction of these reactions would not only improve the quality of life for patients but also save both time and money in regards to pharmaceutical research. Consequently, the drug-gene-pathway (DRUGPATH) meta-database was developed to map known interactions between drugs, genes, and pathways among other information in order to easily identify potential adverse drug events. DRUGPATH utilizes expert-curated sources such as PharmGKB, DrugBank, and the FDA’s NDC database to identify known as well as previously unknown/overlooked relationships, and currently contains 12,940 unique drugs, 3933 unique pathways, 5185 unique targets, and 3662 unique genes. Moreover, there are 59,561 unique drug-gene interactions, 77,808 unique gene-pathway interactions, and over 1 million unique drug-pathway interactions.

## 1. Introduction

The concept of “one disease, one drug” is no longer relevant when applied to complex chronic diseases where drug combination treatments can lead to adverse drug-drug interactions [1,2]. In HIV for example, protease inhibitors such as ritonavir are used but have been known to affect the metabolism of other drugs [3]. To ameliorate adverse interactions such as these, a technique known as targeted drug delivery attempts to reduce the possibility of side effects and other drawbacks of treatment by delivering drugs to specific targets in order to localize their pharmacological activity [2], however, pharmaceuticals tend to be non-specific and on average bind to at least six molecular targets [4], which can lead to off-target interactions that increase the complexity of treatment. That being said, drug repurposing can also be performed where the off-target interactions of a pharmaceutical are utilized to treat diseases not originally targeted. In any case, the ability to predict all interactions of a given drug combination therapy would allow for more effective treatment regimens with fewer side effects, leading to better outcomes towards patient health.

One example of a combination therapy is the use of enbrel and mifepristone as treatment course proposed by our group for Gulf War Illness (GWI) [5,6]. GWI is a chronic multi-symptom illness characterized by gastrointestinal, muscle, and cognitive dysfunction stemming from exposure to battlefield neurotoxins during the 1990–91 Persian Gulf War [5,6,7]. Prior analysis of gene expression in peripheral blood mononuclear cells [8] as well as the dynamic regulatory modeling of the neuro-endocrine-immune response to an exercise challenge [7,9] predicted that an initial inhibition of Th1 cytokines, followed by glucocorticoid receptor (GCR) inhibition, would bring patients from the chronic homeostatic regulation induced by neurotoxin exposure back to a more healthy, stable homeostatic regulatory state [7,9,10]. While subsequent analysis [5] suggested that the best treatment course may be accomplished using enbrel, a tumor necrosis factor alpha (TNFα) inhibitor [11], in combination with the antiglucocorticoid mifepristone [12], great precaution must be used due to the novelty of this treatment regimen as well as the chemical sensitivity found in patients with GWI.

In order to address issues such as this, the drug-gene-pathway (DRUGPATH) meta-database was developed in order to provide a comprehensive mapping of possible interactions between drugs, genes, targets, and biological pathways to inform clinicians of potential adverse effects. The service is accessible at http://drugpath.app. To ensure validity of the resulting interaction database, and to test its functionality, we assessed two drug combinations. The first is the interaction between citalopram and phenelzine. This interaction was chosen as a validation exercise as combining citalopram, a selective serotonin reuptake inhibitor [13], and phenelzine, a monoamine oxidase inhibitor [14], is known to increase the risk of a rare but serious condition called serotonin syndrome, which may include symptoms such as confusion, hallucination, seizure, extreme changes in blood pressure, increased heart rate, fever, excessive sweating, shivering or shaking, blurred vision, muscle spasm or stiffness, tremor, incoordination, stomach cramp, nausea, vomiting, and diarrhea. As such, we expect to find overlap in pathways related to serotonin function as well as pathways related to the resulting symptoms of serotonin syndrome. The second interaction chosen as an exploratory drug combination of enbrel and mifepristone, as it is novel, and relevant to current therapeutic interventions in GWI.

## 2. Results

DRUGPATH amalgamates various expert-curated sources including the Pharmacogenomics Knowledgebase (PharmGKB), DrugBank, the HUGO Gene Nomenclature Committee (HGNC), Guide to Pharmacology (GTP), and the U.S. Food and Drug Administration’s National Drug Code Database (FDA NDC) among others into a single resource available for querying using the graphical interface; see Table 1 for a complete list of all sources used. DRUGPATH contains interactions between drugs, genes, and pathways among other information (e.g., half-life, FDA approval status, indication) in order to easily identify potential adverse drug events (see Table 2 for statistics). Overall, DRUGPATH was developed in order to map the various interactions of a potential drug combination treatment, allowing researchers and clinicians alike to predict as well as mitigate side effects.

The interactions between two sets of drugs, citalopram and phenelzine as well as enbrel and mifepristone, are shown in the following sections using screenshots from the DRUGPATH graphical user interface, which allows users to easily query the DRUGPATH database in order to obtain the drug-gene and drug-pathway interactions (see Figure 1). This interface also allows users to search via gene ensembl, entrez, and symbol identifiers, target names, drug identifiers, indications, and half lives, FDA approval status, and finally pathway names. Any combination of the mentioned options is allowed, so searching for interactions between the TNF gene, “TNF”, and the “signal transduction” pathway for example would be a valid query. Note that the link between a drug and gene is defined as a drug-gene interaction, and likewise, the link between a drug and target or a drug and pathway are drug-target and drug-pathway interactions respectively.

### 2.1. Citalopram and Phenelzine

In this section the results from searching for all interactions between the drugs citalopram and phenelzine are shown. The output contained in each tab, as well as how to interpret it, is explained in each subsection below. Note that searching DRUGPATH via the graphical interface is shown in Materials and Methods.

#### 2.1.1. Pathway Interactions

The pathways shared by both citalopram and phenelzine are shown in the Pathway Interactions tab in Figure 2, where the percentages next to each drug indicates how many of their total genes are associated to the pathway. For example, the “metabolism” pathway has a total of 1024 associated genes, the majority of which (75%) are associated with phenelzine. This type of information is valuable when developing drug combination treatments because potential drug-drug interactions can be identified and mitigated if possible, as in the case of the “signal transduction” pathway that is far more likely to be affected by citalopram than phenelzine. DRUGPATH found a total of 403 shared pathways between citalopram and phenelzine (see Appendix A).

#### 2.1.2. Gene Interactions

Similar to the Pathway Interactions tab, the Gene Interactions tab shown in Figure 3 outputs the genes shared by all drugs. Here, the percentages next to each drug name correspond to how many connections the drug has to the particular gene, or in other words, the strength of association between the drug and gene. The number of connections refers to how many unique drug-gene-target entries there are in DRUGPATH. To illustrate, the cytochrome P450 3A4 (CYP3A4) gene has a total of 1772 connections stemming from citalopram and phenelzine, where each drug is just as likely as the other to affect this particular gene since their percentages are both 50%. On the other hand SLC6A3 is more likely to be affected by phenelzine than citalopram, so if this gene should not be affected during treatment it would be best to find a substitute. DRUGPATH found a total of 59 shared genes between citalopram and phenelzine (see Appendix A).

#### 2.1.3. Network Map

An interactive network map consisting of drug to gene to pathway interactions is shown in the Network Map tab in Figure 4. This was created using visNetwork [26] and igraph [27] with the Fruchterman and Reingold (FR) layout [28], which is a force directed layout where nodes are treated similar to atomic particles; connected nodes will ‘attract’ to one another and be pulled towards a particular direction depending on the number of connections [28,29]. The legend is displayed on the left of the map, where drug nodes are red, genes are blue, and pathways are green. Users can zoom in using either the scrollwheel on their mouse or by clicking the ⊖ and ⊕ buttons on the bottom right side which zooms outwards and inwards respectively. The arrow keys on the bottom left allow the user to move about the map, otherwise the mouse can be used to click and drag as well. When network maps have many drugs, genes, and pathways they can become crowded and difficult to view, so its helpful that clicking on a given node highlights only other connected nodes. For instance, clicking on the “TNF” node will only highlight drugs, genes, pathways directly connected to it; see Figure 5. Alternatively, the drop-down menu located in the top left allows for easy selection of any particular node without having to search and click.

#### 2.1.4. Data

The Data tab shown in Figure 6 allows users to download the raw data corresponding to their search query in comma separated value (CSV) format. This data is pulled directly from the DRUGPATH database and is used to produce the information shown in all other tabs. In this tab users are able to scroll through all of the data, and to download they can simply click the “Download” button located in the top left.

### 2.2. Enbrel and Mifepristone

Similar to above with citalopram and phenelzine, this section displays the pathway and gene interactions between the drugs enbrel and mifepristone.

#### 2.2.1. General Overview

Overall, there were a total of 52 genes and 61 pathways between enbrel and mifepristone, with the progesterone receptor (PGR) gene and the immune system pathway having the most associations. See Figure 7. These results are consistent with the known function of these drugs.

#### 2.2.2. Pathway Interactions

Of the 61 pathways shared between enbrel and mifepristone, the first few pathway interactions are shown in Figure 8. The full results (see Appendix A) show many immune-related pathways such as Th17 cell differentiation, interleukin (IL) 2 (IL-2), IL-4, and IL-13 pathways among many others. As enbrel targets the immune pathways this is a consistent result. Beyond this there are also pathway interactions involved with leptin signaling, gastrin, and ghrelin, which suggests a potential effect on appetite and digestion. Additionally, there are interactions in adipogenesis, adipocytokine and visceral fat deposit pathways suggesting a potential effect on weight gain or loss.

#### 2.2.3. Gene Interactions

While all 61 pathways associated with enbrel and mifepristone were affected by both drugs, none of the 52 total genes are shared, or interact, between these drugs. In this case, a message of ‘None available’ appears in the Gene Interactions tab (see Figure 9).

## 3. Discussion

DRUGPATH is a meta-database that consolidates expert-curated sources in order to map the interactions between drugs, genes, targets, and pathways, of which there are currently ≈60,000 drug-gene interactions, ≈94,000 gene-pathway interactions, and over 1.2 million drug-pathway interactions. In addition to all of the sources used in DRUGPATH, there are numerous other freely available biological, pharmacological, genetic, and even interaction databases, however, to our knowledge none of these other databases currently provide drug-gene, drug-pathway, and gene-pathway interactions directly. While DrugBank, PhID, and others provide drug-drug interaction checkers on their websites, none goes so far as to provide both the pathways and genes predicted to be involved with a given interaction.

For instance, Figure 2, Figure 3, Figure 4, Figure 5 and Figure 6 show the predicted results of the drugs citalopram, a selective serotonin reuptake inhibitor [13], and phenelzine, a monoamine oxidase inhibitor [14]. These two drugs in combination can induce serotonin syndrome [30], but services such as DrugBank and RXList only state at most whether an interaction occurs and its effects, but not the genes and pathways involved. DRUGPATH on the other hand finds 481 shared pathways and 59 shared genes between citalopram and phenelzine, which includes serotonin related pathways such as “selective serotonin reuptake inhibitor pathway” and “serotonergic synapse” pathway, but also dopaminergic-involved pathways (e.g., “dopaminergic synapse pathway” and “naltrexone action pathway” [31]) as well as “androgen and estrogen biosynthesis and metabolism pathway” and “codeine and morphine metabolism pathway”, which have been verified in literature to be involved with both citalopram and phenelzine’s pharmacological actions [32,33,34,35,36]. These findings lend confidence in DRUGPATH’s ability to identify potential reactive drug combinations.

The combination of enbrel and mifepristone in Figure 7, Figure 8 and Figure 9 were performed to demonstrate the predictive capabilities of DRUGPATH since there is currently no literature available documenting the interactions between these two drugs. Here, the major hypothesis of GWI pathophysiology involves the combination of stress and exposure to battlefield neurotoxins triggering a neuroinflammatory cascade, leading to altered homeostatic regulation [5,6,7]. Our computational models have predicted the inhibition of Th1 inflammatory cytokines (e.g., TNFα) followed by inhibition of the GCR would return the patient back to stable homeostatic regulation [7,10]. Such models indicated the use of enbrel and mifepristone. Enbrel is a TNFα inhibitor FDA approved to treat arthritis and plaque psoriasis, with adverse interactions including headache, infection, rhinitis, and sepsis [11,37,38,39]. Mifepristone is a high affinity progesterone and glucocorticoid receptor antagonist used for pregnancy abortion, with adverse effects such as diarrhea, fever, cramping, nausea, and vomiting [12,40,41]. DRUGPATH predicted that these two drugs would interact via immune pathways, such as the aptly named “immune pathway” affected by almost all genes connected to enbrel and the ELANE gene from mifepristone. This interaction suggests that these drugs in combination would likely affect immune functioning, which aligns to the proposed treatment strategy from our group of enbrel and mifepristone for GWI [5,7,10]. This is consistent with the expected effect of these drugs, and would normally be monitored when undertaking these drug treatment courses. The identification of overlap in effects on leptin signaling, gastrin, and ghrelin pathways suggests a potential effect on appetite and digestion, which suggests additional monitoring is warranted of digestion in patients undergoing a combined enbrel-mifepristone treatment course. Likewise interactions in adipogenesis, adipocytokine and visceral fat deposit pathways suggests increased monitoring of a patients weight gain or loss during this drug combination.

While analysis of these two drug combinations shows consistency with known drug effects, and shows the potential of DRUGPATH to identify potential adverse drug effects, DRUGPATH’s greatest strength is also its biggest limitation. Namely, it relies heavily on the information from the sources used in its curation. While curation performed by experts is often a favorable trait for biomedical databases, it can lead to errors when dealing with thousands to millions of entries. That being said, sources often make corrections with every new update of their database, which is why we established the requirement that a source must be actively maintained in order to be incorporated into DRUGPATH. Additionally, the architecture of DRUGPATH where each source is self-contained in their own sub-directory was a deliberate choice, since it allows users to not only enable/disable specific sources at will, but it makes applying and checking for updates more convenient and streamlined. Furthermore, DRUGPATH currently does not measure the strength of association between a gene and drug for instance, which is due to the fact that it can only provide data included in each database. While all sources contribute incredibly valuable information, experimental data such as binding energies are unfortunately not provided.

## 4. Materials and Methods

DRUGPATH was built using MATLAB versions 2017b and 2018a, refs. [42,43] as well as Python version 3.5 [44], and is fully modular in nature. This means that each source is confined to their own sub-directories where the raw input file and all scripts required for pre-processing, data extraction, and amalgamation to the DRUGPATH dataset are found. We refer to each source’s sub-directories as ‘plugins’ because they can be independently modified without affecting any other source or editing the existing codebase. Not only that, but each source can be easily enabled or disabled using a plaintext configuration file. DRUGPATH was saved in the SQLite database file format, which is lightweight, cross-platform, and very fast for querying millions of entries compared to other formats such as CSV files or Microsoft Excel, making it accessible and easy to use [45].

Generating DRUGPATH consisted of first utilizing PharmGKB [22] to obtain drug-gene interactions, generic and proprietary drug names, and drug identifiers for services such as ChemSpider [46], Uniprot [47], ChEMBL [48,49,50], and so on. Next, drug-target interactions, additional drug identifiers, and half-life information were obtained from both DrugBank [17,18] as well as T3DB [24,25]. GTP [20] provided even more drug names and identifiers, and HGNC [21] added missing gene ensembl, gene entrez, and gene symbol identifiers for entries. Gene-pathway interactions were mapped using CPDB [15,16], and repoDB [23] provided drug indications. Finally, the database was formatted so that each entry contains only one drug to gene to target to pathway interaction.

One of the major issues when amalgamating many databases is redundancy. Here, targets such as IL-2 can have many different names such as “interleukin-2”, “IL-2”, “interleukin two”, “IL2”, “interleukin 2”, and so on, while drugs can have a variety of generic as well as proprietary/trade names. To address this, the various drug identifiers used by each source including DrugBank, PubChem substance, PubChem compound, DrugBank, ChEBI, etc. identifiers were all saved in the DRUG_ID column. This allowed DRUGPATH to map drugs by not only their names, but also their identifiers as well. For instance “mifepristone” (generic name), “RU-486” (generic name), “mifepristona” (Spain), “mifepriston” (Germany), “mifépristone” (France), “mifegyne” (trade name), etc. were all associated with one another since they share the same DrugBank identifier. It’s important to note that all of the unique drug names were saved in the DRUG_NAME column so that users could easily search a given drug and DRUGPATH would return the appropriate results. In regards to combating redundancy in targets, the gene entrez, gene ensembl, and gene symbol identifiers, found in columns 1–3 in DRUGPATH respectively, were used in the same manner as drug identifiers to match targets that otherwise have non-matching names.

### 4.1. Fetching and Updating

Before a new source can be added to DRUGPATH, it is first vetted against current literature as well as the other sources used in the dataset in an attempt to verify if the data is accurate. Furthermore, the source needs to be actively maintained where it’s regularly receiving updates in order to be considered for inclusion, otherwise the likelihood of mapping false correlations from outdated information increases. That all being said, this vetting process is complicated due to a variety of factors, one of which includes misspellings of drug, gene, and/or target names within sources that are otherwise trustworthy. For example, the FDA NDC database contains information on all FDA approved drugs such as proprietary/generic brand names, but there were a few instances of misspelled drug names which made tasks such as mapping drugs to genes more arduous. Additionally, all of the sources that make up DRUGPATH contain thousands of entries, making detection of outdated and/or incorrect entries a very difficult task. Fortunately, these erroneous entries are often quickly corrected when reported, and they only make up a very small share of the entire database.

Updating is completed through a Python script that first downloads the source from its website into a temporary directory, performs a backup of the source’s existing database within the plugin directory, and then moves the updated source file into the plugin directory.

### 4.2. Plugins

Every source is contained in their own plugin directory, which includes the scripts for preparation and amalgamation. Here, the preparation script extracts only relevant information, such as data on humans rather than animals and obtaining universal drug identifiers from ChEMBL or Uniprot instead of the database’s internal identifiers, and normalizes naming conventions (e.g., TNF-α vs TNFa vs TNFα) to be compatible with the existing DRUGPATH entries. This extracted data is saved to a SQLite database file format, and the amalgamation script queries this SQLite database in order to merge it with the existing DRUGPATH database. See Figure 10 for a graphical overview of how the plugin directories are structured.

### 4.3. Columns

There are a total of 10 columns within the DRUGPATH database, the details of which are explained below. As an overview, column 1 contains gene entrez identifiers, column 2 is comprised on gene ensembl identifiers, and gene symbols are located in column 3. There are various gene identification systems available, but these three were the most ubiquitous so they were all used. Column 4 contains the targets associated with the genes in previous columns (e.g., gene symbol “CYP3A4” corresponds to target “cytochrome P450 3A4”), and are used for drug-target interactions. Numerous identifiers for each drug are available in column 5, which directly contribute to all of the drug names found in column 6 by allowing different names to be mapped together across distinct sources. Column 7 includes the half-lives for as many drugs as possible, and column 8 contains the FDA status for all drugs found within the appropriate sources. Lastly column 9 is comprised of the pathways associated with each gene, and column 10 contains the FDA indications for the drugs.

The DRUGPATH SQLite database file contains 3 separate tables: main, drug_id, and drug_name. The 10 columns described above all exist in the main table, however, once DRUGPATH has been generated, one of the final post-processing steps is to replace both the drug identifiers in columns 5 and the drug names in column 6 with the same integer. This integer serves as an identifier within both the “drug_id” and “drug_name” tables, which contain the numerous identifiers as well as generic/trade names for the given drug respectively; see Figure 11. This procedure eliminates repeating the same long string of drug identifiers and names for multiple entries, which not only reduces the total size of the DRUGPATH SQLite file, but also makes querying the database faster and more efficient.

#### 4.3.1. Gene Entrez, Ensembl, and Symbol Identifiers

Columns 1–3 of the DRUGPATH database contain gene identifiers for the genes that code for their corresponding targets in column 4. While there are various gene identifier formats that exist, DRUGPATH uses only three of the most prominent: the National Center for Biotechnology Information’s (NCBI) gene entrez, the European Bioinformatics Institute (EMBL-EBI) and the Wellcome Trust Sanger Institute’s gene ensembl, and finally HGNC’s gene symbols. All of the drug-gene relationships found in DRUGPATH are provided by PharmGKB, which builds its database using both manual curation as well as natural language processing techniques to extract this information from published pharmacogenetics and pharmacogenomics studies [22].

#### 4.3.2. Target

The targets in column 4 were provided for drug-target interactions, and are among the most volatile data in DRUGPATH due to the fact that so many targets have numerous names. To illustrate, the ‘Glutamate receptor ionotropic, NMDA 1’ and ‘Glutamate Ionotropic Receptor NMDA Type Subunit 1’ are two different names for the same target, and unlike drug names, there are not many sources available that map these target names together. For this reason target-pathway interactions were actually performed using gene symbols instead, where ‘GRIN1’ corresponds to both of the glutamate receptor names from the previous example.

#### 4.3.3. Drug Identifiers

The issue of various names corresponding to the same target occurs with drugs as well, where there are many generic as well as proprietary/trade names for a drug. This was ameliorated in part by the DrugBank, T3DB, and other sources that provided the names, but the bigger solution was to utilize unique drug identifiers from well established sources such as ChEMBL, ChemSpider, and Uniprot among others which represents all of a drug’s generic and trade names. However, unlike the gene identifiers where a selected few formats are most commonly used, a drug identifier may be available in one database but not in another, so once a new identifier was found within a source it was added to the existing drug identifiers located in column 5, creating an encompassing list of associated identifiers for a given drug. Figure 11 shows that this column is comprised of integers which correspond to the actual drug identifiers found in a separate “drug_id” table. In essence, a single number can represent numerous drug identifiers which saves space and speeds up user queries by reducing the amount of text to search through.

#### 4.3.4. Drug Name

Column 6 contains all of the drug names such as generic and trade names for each drug. Similar to the process used for drug identifiers, once a new name for a given drug was found it was automatically added to the list of drug names within this column, which improves the mapping of drug-gene and drug-pathway interactions as more and more databases are incorporated into DRUGPATH. As more associated names for a drug are linked to together, the accuracy and reliability of our network increases. Similar to column 5, the drug names in this column are replaced by a single integer, which serves as an identifier in the “drug_name” table that corresponds to all of the possible generic/trade names for the given drug.

#### 4.3.5. Half Life

Column 7 contains the half-live information for as many drugs as possible. This data was sourced from DrugBank and T3DB, meaning that only drugs found in these two databases will have associated half-life information.

#### 4.3.6. Status

The FDA status for each drug is found in column 8 and can be one or more of: *approved*, *experimental*, *investigational*, *nutraceutical*, *illicit*, *vet approved* and/or *withdrawn*. Drugs are able to have one or more different statuses because they can be approved to treat one disease and in trial for approval to treat another (status: *approved* and *experimental*), may be approved for both humans and animals, such as Butorphanol (status: *approved* and *vet approved*), and so on. Similar to the half life column above, if the drug was not found in the source providing the information, in this case the FDA NDC database, then the drug will not have a status.

#### 4.3.7. Pathway

CPDB [15,16] provided the gene-pathway interactions found in column 9.

#### 4.3.8. Indication

Finally, column 10 contains each drug’s indication, or in other words, what disease or condition is being treated by this particular drug. DrugBank, T3DB, and repoDB were all utilized to provide this data.

### 4.4. Graphical Interface

Once DRUGPATH has been successfully generated, code written in the R programming language [51] was used to quickly and easily query the database to obtain interactions; see Figure 1. The graphical frontend was created using Shiny [52,53], which passes the user input to the backend R script in order to produce the gene and pathway interactions, as well as a network map that models the drug, gene, pathway interactions. On the hompage, the user can input drug names, gene and drug identifiers, half life information, status, and pathway names in order to search DRUGPATH. Note that any combination of drugs, identifiers, etc. can be queried at once. This graphical interface is located at the web address http://drugpath.app.

#### 4.4.1. Searching

To query the database, input a comma-delimited list of search terms (e.g., “query1, query2, query3, etc.”) into the search bar, and check all categories that the search terms belong to. For example, to find interactions for the drugs “citalopram” and “phenelzine”, the search box would contain these two terms, separated by a comma, and the checkbox for “Drug Name” would be checked. As search terms are entered, each individual term will be shown at the bottom of the page; see Figure 12. Additionally, the submit button is only shown once a search term has been entered and at least one column has been selected.

#### 4.4.2. General Overview

Once the user has submitted his query, four new tabs are shown. The first of which is the General Overview tab (see Figure 13), which displays a quick synopsis of the genes and pathways associated with the search query. Continuing with the citalopram and phenelzine example, it is shown that there are a total of 59 genes and 471 pathways associated between these two drugs. The “Most Associations” metric refers to the gene or pathway with the most amount of connections, defined as the total number of unique drug-gene-target entries in DRUGPATH for the genes, and the total number of unique drug-gene-target-pathway entries for the pathways. Essentially, this is a measurement of how prominent a given gene or pathway is within an interaction network. In this case the “SLC6A4” gene and the “metabolism” pathway had the most connections between citalopram and phenelzine in this particular network.

## Figures and Tables

**Figure 1 ijms-21-03171-f001:**
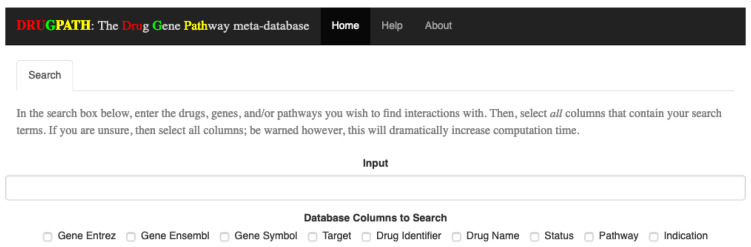
DRUGPATH Graphical User Interface.

**Figure 2 ijms-21-03171-f002:**
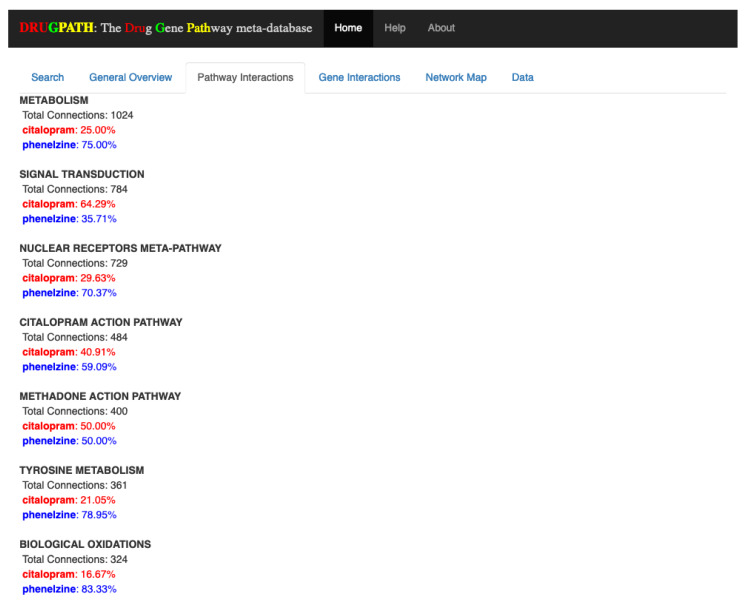
Pathway interactions between citalopram and phenelzine. Note that only the first few results are shown.

**Figure 3 ijms-21-03171-f003:**
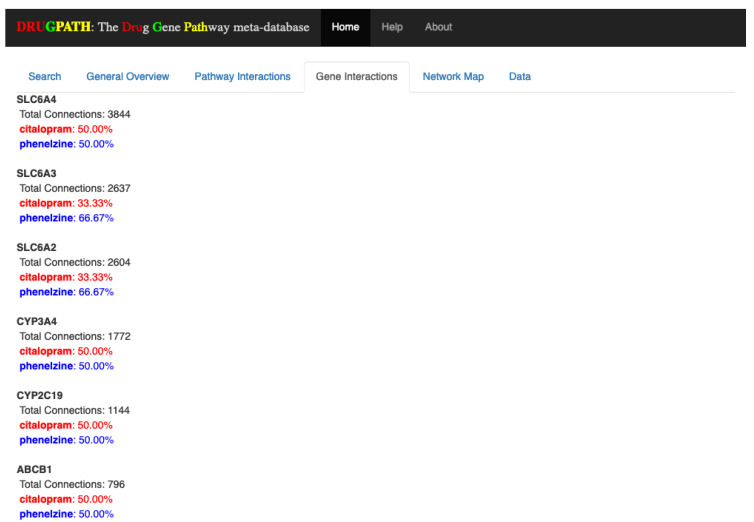
Genes interactions between citalopram and phenelzine. Note that only the first few results are shown.

**Figure 4 ijms-21-03171-f004:**
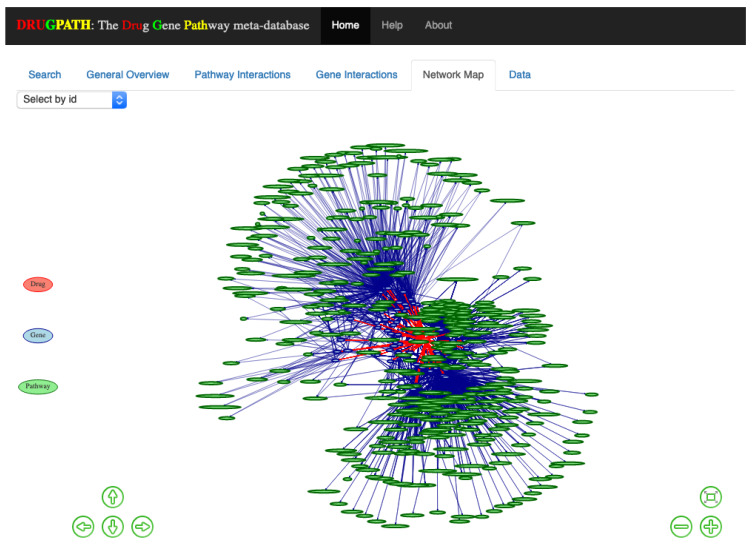
Overview of all the genes and pathways associated with citalopram and phenelzine.

**Figure 5 ijms-21-03171-f005:**
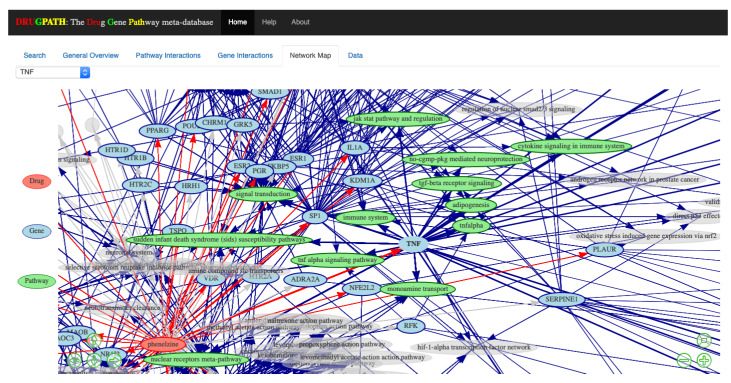
Closer examination of all drugs and pathways associated with the gene **TNF**. Any drug, gene, or pathway not associated with this gene are greyed out and made transparent.

**Figure 6 ijms-21-03171-f006:**
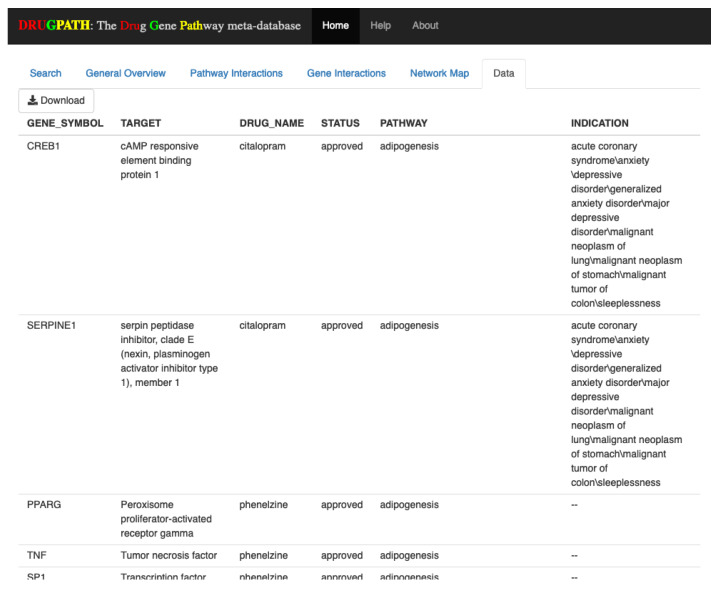
Data available for download.

**Figure 7 ijms-21-03171-f007:**
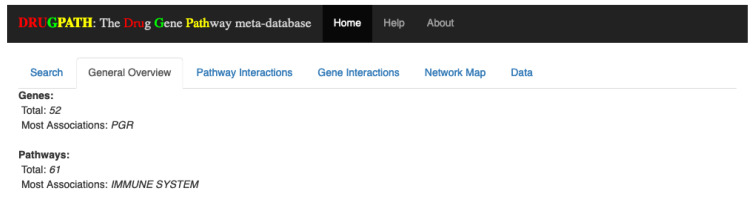
General overview of the interacting genes and pathways between enbrel and mifepristone.

**Figure 8 ijms-21-03171-f008:**
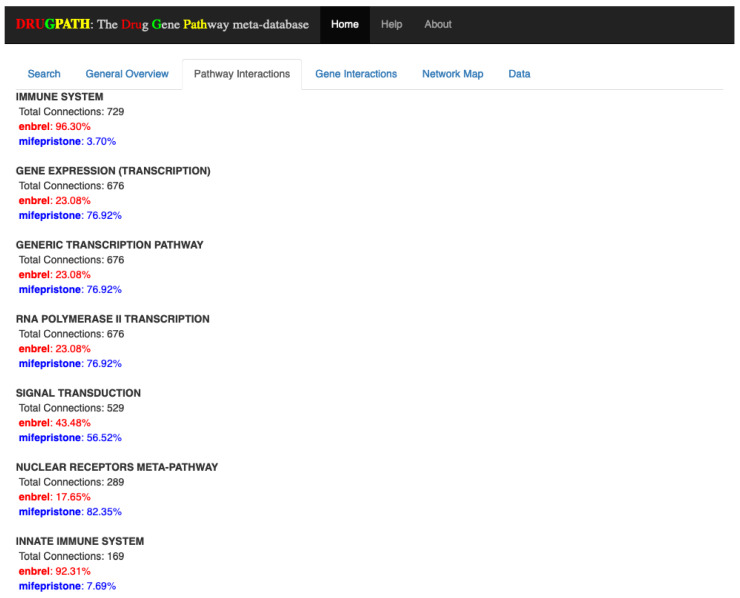
First few pathway interactions between enbrel and mifepristone.

**Figure 9 ijms-21-03171-f009:**
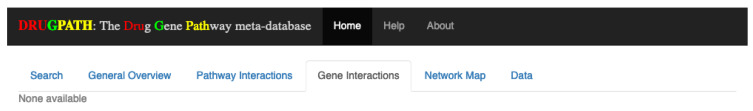
Message that appears when no gene interactions are found.

**Figure 10 ijms-21-03171-f010:**
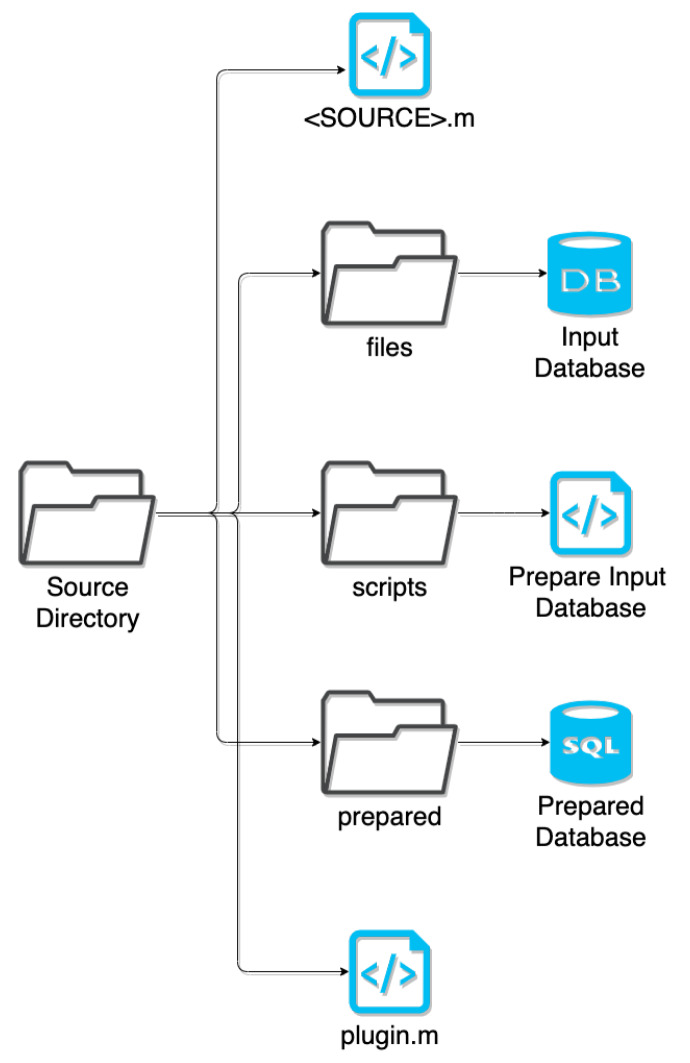
Structure of each source’s sub-directory. **<SOURCE>.m** refers to the MATLAB script used to amalgamate the **Prepared Database** with the existing DRUGPATH database. **plugin.m** is called by the main MATLAB script that generates the database, and its job is to execute the script that prepares the raw input database as well as execute **<SOURCE>.m**.

**Figure 11 ijms-21-03171-f011:**
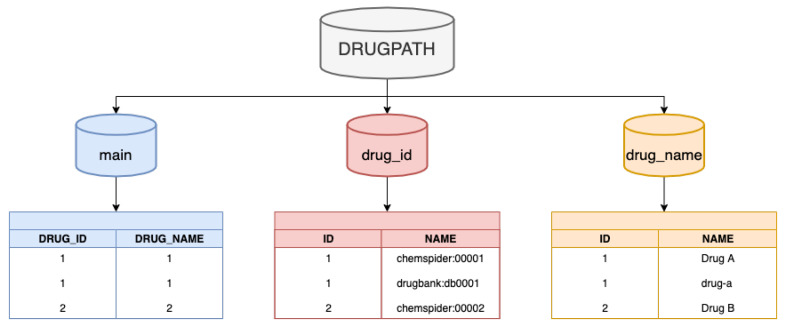
Structure of the DRUGPATH SQLite database, where **main**, **drug_id**, and **drug_name** are all separate tables within the file. Here, DRUG_ID “1” corresponds to multiple identifiers within the **drug_id** table for DRUG_NAME “1”, which in turn represents the various names of Drug A within the **drug_name** table. Note that within the **main** table, there are a total of 10 columns, but only the DRUG_ID and DRUG_NAME columns are shown in this figure.

**Figure 12 ijms-21-03171-f012:**
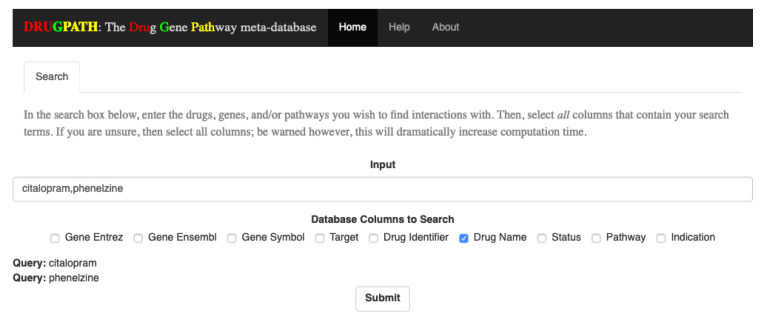
As the user inputs the search terms, they are shown at the bottom of the page.

**Figure 13 ijms-21-03171-f013:**
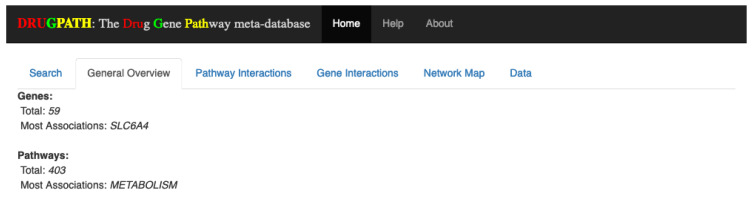
Quick overview of the total number of genes and pathways found, as well as the gene and pathway that had the most connections.

**Table 1 ijms-21-03171-t001:** The sources and their roles within the DRUGPATH database.

Source	Role
ConsensusPathDB (CPDB) [15,16]	pathways, pathway identifiers, gene symbols
DrugBank [17,18]	gene symbols, drug-target interactions, drug names, identifiers, half-lives, FDA approval status, and indications
FDA NDC [19]	drug names, FDA approval status
Guide to Pharmacology (GTP) [20]	drug names and identifiers, gene symbols, and targets
HGNC [21]	gene entrez, ensembl, and symbol identifiers
PharmGKB [22]	drug names, drug-gene interactions
repoDB [23]	drug names and indications
Toxin and Toxin-Target Database (T3DB) [24,25]	gene symbols, drug names, identifiers, half-lives, FDA approval status, and indications, drug-target interactions

**Table 2 ijms-21-03171-t002:** DRUGPATH’s statistics.

Category	Available
Drugs	12,940
Genes	3662
Targets	5185
Pathways	3933
Drug-Gene Interactions	59,561
Drug-Pathway Interactions	1,053,633
Gene-Pathway Interactions	77,808

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
