# Peer review of "DRUGPATH: The Drug Gene Pathway Meta-Database"

_ijms, 2020, doi:10.3390/ijms21093171_

Round 1

Reviewer 1 Report

Dear Editor,

First of all, I would like to thank you for giving me the opportunity to review the manuscript “DRUGPATH: The Drug Gene Pathway Meta-database” by R. Jaundoo et al.

The authors present a new meta-database to help prediction of drug effect. The tool is freely distributed, and can be useful for the community.

The manuscript is globally well written.

I have a few minor concerns regarding the manuscript.

Minor concerns:

  • Authors should clearly state how they define what is a ‘link’ between a drug and a target, gene or pathway.
  • Authors could rapidly describe the architecture of the database. Authors mention the existence of 10 columns. Does this mean that there is one table with 10 columns, or are these column distributed over several tables?
  • Page 2, line 37. “regiem” -> “regime”
  • Page 2, line 59. “drug events see table 2 for statistics.” -> “drug events (see table 2 for statistics).”
  • Page 9, line 167. “DRUPATH” -> “DRUGPATH”
  • Page 11, line 264. “where a select few formats” -< “where a selected few formats”
  • Page 13, line 309. “Once the user has submitted their query” -> “Once the user has submitted his query”

Sincerely,

Author Response

We thank the reviewer for their comments.

In general, all spelling and grammatical errors pointed out by each reviewer were fixed as below.

Responses to Reviewer 1 are as follows.

=======
Results
=======
Table 1: Added citations for each of the sources in the "Source" column.
Lines 69-71: Addressed reviewer 1's comments of what a link between a drug and gene/target/pathway is.

=====================
Materials and Methods
=====================

Lines 262-269 & Figure 11 & 296-299 & Lines 306-308: Addressed reviewer 1's comments regarding the architecture of the database.

Reviewer 2 Report

Manuscript Title:

DRUGPATH: The Drug Gene Pathway Meta-database

Manuscript ID : ijms-746469

Recommendation: Publish in Int. J. Mol. Sci after minor revisions

Comments:

Exploring drug-gene, drug-pathway, and gene-pathway interactions is critical to understand drug adverse effect, drug-drug interaction, and drug repurposing. This paper attempts to provide a solution by creating a comprehensive drug-gene-pathway meta database, called DRUGPATH. DRUGPATH was built based on the curation of a couple of public database with a GUI implementation, and is free for download. I believe this tool/database would be useful in different phases of drug discovery. However, some concerns need to be addressed before publication in Int. J. Mol. Sci.

  1. How to deal with redundant information from different public sources?

DRUGPATH integrated a variety of available database as shown in Table 1. I believe there must exist redundancy and mismatch for the information of drugs, genes, and pathways. How did the authors clean and merge the database together?

  1. How to define a drug-gene association and its strength?

I wonder how to define a drug-gene association in DRUGPATH when curating the data. What types of experimental data and the cutoffs were used to define a drug-gene association? If there were multiple experimental data associated with a drug-gene association from one or more database, how to unify them? Similarly, is it possible to define the strength for each individual drug-gene association in DRUGPATH?

  1. Does DRUGPATH has a webservice instead of installing the package locally? As such, the readers can access and try DRUGPATH more easily.

  1. In Table I, it would be good to add another column to show the link or at least references for each source database.

  1. In Line63, it should be "DRUGPATH" instead of "DRUGBANK".

Author Response

We thank the reviewer for their comments.  

In general, all spelling and grammatical errors pointed out by each reviewer were fixed as outlined below.

The revisions made at Reviewer 2's requests are as follows.

==========
Discussion
==========
Lines 188-192: Addressed reviewer 2's comments in regards to providing the strength of association between drugs and genes.

=====================
Materials and Methods
=====================
Lines 209-219: Addressed reviewer 2's comments in regards to how DRUGPATH handles redundancy.

Lines 267-270: Addressed reviewer 2's comments in regards to the statement "I wonder how to define a drug-gene association in DRUGPATH when curating the data"

Reviewer 3 Report

The authors present a novel meta-database that incorporates drug, gene, and pathway interactions towards the prediction of drug-drug interactions (DDI) that result in adverse events (ADE). The generation of this meta-database is done in a very sophisticated way: using a collection of publicly available and well-curated databases for each component of DRUGPATH (drug-gene, gene-pathway, etc.). Moreover, using all of these aspects for the prediction of DDI is important, which is made clear by the authors in the first sentence of the introduction: "The concept of “one disease, one drug” is no longer relevant when applied to complex chronic diseases where drug combination treatments can lead to adverse drug-drug interactions".

There appears to be a high level of curation for maintaining this meta-database  which gives credence to predictions being made. Additionally, the authors make a point to discuss the reliance of DRUGPATH on these various sources and the associated fidelity of all sources. The authors take care to also describe the vetting process of each source, which should mitigate any issues that could arise from unreliable databases.

That all being said, I have two concerns, though minor. The first being the inability to directly predict DDI for two drugs. There is a high granularity to this data, allowing for the determination of specific genes and pathways that two drugs interact with, thus leading a user to consider certain things when co-prescribing. For example, the authors write "The identification of overlap in effects on 174 leptin signaling, gastrin, and ghrelin pathways suggests a potential effect on appetite and digestion, which suggests additional monitoring is warranted of digestion in patients undergoing a combined etanercept-mifepristone treatment course." Again, the level of granularity is great to identify these specific pathways overlapping for these two drugs, however the inability of this meta-database to actually predict a confidence for a DDI and/or a specific resulting ADE is a shortcoming. The capability to directly produce a confidence score for which the interaction between two drugs will result in an ADE would be a very helpful tool for many different users. In addition, this type of prediction can be compared against known DDIs from sources such as DrugBank, which the authors do mention, to quantify the accuracy of the meta-database DDI predictions.

My second concern is the lack of a live server for end-users. Though this is not necessary, a common end-user may be limited in their computational skills, making the downloading and installing various dependencies a large barrier to using DRUGPATH. Therefore, an easy to access web front-end would enable many more users to use DRUGPATH.

Overall, this manuscript is well written and thoroughly explains the meta-database, DRUGPATH. I recommend this paper be accepted in its current form, while taking my considerations into account for future development.

Author Response

We thank the reviewer for their comments.

We are in the process of addressing the webserver issue.  We have secured the URL http://drugpath.app and have implemented the front end.  We are currently beta testing the service, but in the interest of time we have submitted these revisions while these tests are ongoing.  The service will be available to coincide with publication if approved.
